# Barriers to and Facilitators of Providing Care for Adolescents Suffering from Rare Diseases: A Mixed Systematic Review

Pelagia Tsitsani [1,2,*], Georgios Katsaras [2] and Elpidoforos S. Soteriades [1]

1  Healthcare Management Program, School of Economics and Management, Open University of Cyprus, 2220 Nicosia, Cyprus; elpidoforos.soteriades@ouc.ac.cy

2  Paediatric Department, General Hospital of Pella—Hospital Unit of Edessa, 58200 Edessa, Greece; gkatsaras84@gmail.com

*  Correspondence: pelazina@gmail.com; Tel.: +30-693-665-5540

**Abstract:** Rare Diseases (RDs) in adolescents are characterized by low frequency and clinical heterogeneity, are chronic and deliberating and demand a multidisciplinary approach as well as costly and specialized treatments. Comprehending patients' and parents' needs through a mixed systematic review is essential for healthcare system planning. This mixed systematic review explored barriers to and facilitators of effective care for adolescents with RDs as perceived by patients and their parents. Three databases (2008–2023) were searched and twenty-five articles were selected and critically appraised with the Mixed Methods Appraisal Tool (MMAT; version 2018). The review followed a convergent integrated approach for data extraction according to Joanna Briggs Institute's (JBI) mixed method systematic review (MMSR) methodology. Subsequent reflexive thematic analysis categorized the barriers and facilitators into five levels (intrapersonal, interpersonal, institutional, community and public policy) following the conceptual framework of the socio-ecological model for healthcare promotion. Barriers on the institutional and public policy level stood out as the most frequently reported, resulting in unmet care needs and frustrating family dynamics. National and regional healthcare policies are rarely actually linked to pragmatic intervention programs with a measurable impact on patients' welfare. Targeted strategies involving primary care infrastructure and personnel for better coordination and management of the disease are discussed.

**Keywords:** Rare Diseases (RDs); adolescents; healthcare provision; mixed systematic review

## 1. Introduction

Rare Diseases (RDs) affect 3.5–5.9% of the population, with 80% considered genetic and approximately 70% primarily present in childhood (0–18 years) [1]. RDs have, by definition, a difficult and prolonged time until diagnosis, a chronic-degenerative course with dubious outcomes, and affect young patients who require expert and expensive treatments (e.g., high-cost drugs and/or technology machinery) [2,3]. Therefore, suffering adolescents are confronted with complex medical needs that demand a multidimensional milieu of care. Psychosocial developmental considerations, school attendance, mental health surveillance and transition to adult services have to be integrated into the patient's management plan [4]. However, the adolescent population is seldom examined exclusively apart from adult patients [5].

The diversity of healthcare systems makes a universal approach towards RDs difficult, although the need for worldwide policies and comprehensive action plans is well established [6]. National plans forged by developed countries mainly focus on creating National Rare Disease Patient Registries, organizing and institutionalizing Specialized Centers of Reference and Excellence, as well as legislating frameworks for orphan drug development and approval [7]. However, patients worldwide still experience administrative obstacles in accessing public health and social services and are often driven to catastrophic

expenditure [8,9]. The health-related quality of life (HRQL) of the affected adolescents is characterized by limitations in functionality and physical activity due to the disease itself, school absenteeism, problematic integration into peer groups (social stigma) and frustration due to sophisticated therapies as well as the uncertainty of the prognosis [10]. The academic literature records the RDs' negative impact on the whole family, especially in cases of the frequent hospitalization or deterioration of the sick family member [11]. Parental quality of life (QL) is also severely compromised with a lack of personal time, the significant disruption of professional and social relationships, and physical and mental hardships. Parents are encumbered with the additional role of organizing the co-management of the disease between various specialties and at different levels of care, reporting disparities in accessing medical, social and psychological care [12]. In most cases, the provision of care is fragmented, incomprehensive and circumstantial, leading to mistrust in healthcare systems and relevant authorities [13].

Nevertheless, few studies emphasize the problems of healthcare provision as a core element of eliminating the family's burden of the disease [14]. Therefore, the aim of this study is to address this gap by conducting a mixed systematic review on the perceptions of adolescent patients with RDs and their parents regarding healthcare provision. Moreover, we intend to identify the barriers and facilitators of effective care and finally suggest targeted strategies for improvement.

## 2. Materials and Methods

### 2.1. Study Design

A mixed systematic review was structured based on Joanna Briggs Institute's (JBI) mixed method systematic review (MMSR) methodology [15]. The JBI Summary software and online tools were selected in order to guide and support the entire review process [16].

A mixed systematic review is chosen over a "classical" systematic review when difficult, sensitive and multifactorial issues are being investigated. The synthesis of qualitative, quantitative and mixed studies is considered essential when there are interconnections and overlaps among different dimensions of the problems identified [17]. In order to optimize healthcare management, the incorporation of young patients' and families' narrations of navigating the healthcare system proves to be of the utmost value. The mixed method approach usually deepens into assisting healthcare decisions and policymaking by establishing "barriers and facilitators" together with capturing personal perspectives and experiences [18].

Therefore, the MMSR research approach was considered appropriate in order to adequately address the multi-faceted subject of our review.

### 2.2. Review Questions

The research questions of this review were:

1.  What are the perceptions of adolescents with RDs about the provision of healthcare?
2.  Are they satisfied, or do they recognize barriers and facilitators?
3.  What do their parents think of the existing healthcare provision?
4.  What do the parents view as barriers and facilitators?
5.  Which healthcare strategies and/or interventions are reported to have a positive impact on the quality of life (QL) and health-related quality of life (HRQL) of adolescent patients and their parents?

### 2.3. Inclusion Criteria

This study acknowledged the PICO (population, phenomena of interest and context) framework, as follows:

- Population: Adolescents suffering from Rare Diseases with an established diagnosis and their parents. There were no geographical or socioeconomic limitations.
- Phenomena of interest: Barriers and facilitators regarding healthcare provision.

- Context: All aspects of healthcare provision (structures, processes and outcomes) in all healthcare settings (inpatient, outpatient, primary, secondary or tertiary).

The selected studies followed the following inclusion criteria:

- Peer-reviewed English publications.
- Original papers.
- Participants included adolescents and their parents describing experiences in the healthcare system.
- Qualitative, quantitative or mixed methodologies.
- Potentially relevant to the research questions.

The exclusion criteria were limited to (a) non adolescents (19+ years), (b) language (other than English), (c) any articles published prior to 2008, (d) any review articles and (e) unavailability of full-text articles.

### 2.4. Definitions

According to Bach-Mortensen and Verboom [19], most mixed methods studies do not explicitly define the terms "barriers and facilitators" and this may reflect negatively on mapping the importance of these factors. Therefore, in this review, we identified "barriers and facilitators" as used by the Integrated Checklist of Determinants of Practice (ICDP): "being factors that might prevent (barrier) or enable (facilitator) improvements in healthcare provision for the targeted population" [20]. According to Bach-Mortensen and Verboom, "barriers and facilitators" studies should also clearly state the way specific factors are extricated and subsequently amalgamated as well as describe all of the steps towards this synthesis [19].

The review employed the World Health Organization's (WHO's) definitions for RDs: "diseases with a frequency of less than 6.5 to 10 per 10,000 people" [21] and adolescence: "the phase of life between childhood and adulthood, from ages 10 to 19" [22]. Adolescence is the period between 10 and 19 years old. It is a unique era of human development where multifaceted physical, mental and cognitive evolution occurs and the time during which healthy hygiene and self-care patterns are predicated and evolved in adult life. The transition from childhood to adult services for individuals with chronic diseases usually takes place between 16 and 19 years old, that is, during middle and late adolescence, and is highly dependent on the nature of the healthcare system and the family's preferences [22,23].

### 2.5. Conceptual Framework

The conceptual framework of this review adopted the ecology perspective for healthcare promotion as previously developed in the work of the researchers Uhm and Choi for chronic diseases [18]. The writers taxonomized the barriers and facilitators using five levels: intrapersonal factors, interpersonal processes, institutional factors, community factors and public policies. This model arranges consensus of understanding the interactive views and experiences of patients' and families' scopes as well as systemic/organizational and environmental parameters.

Intrapersonal factors include the characteristics of the individual such as self-esteem, developmental history, knowledge and attitudes. Interpersonal factors include engaging in social and personal relationships such as interactions among family, school and work. These factors also contain communication obstacles among patients, caregivers and healthcare providers. The institutional level contains healthcare organizations with certain functional rules and administrative capabilities; therefore, this level reflects the governmental or non-governmental organizations' efficiency of performances and implicates financial issues [24]. Community factors deal with local healthcare facilities and support systems [25]. The public policy level refers to the laws, regulations, actions and decisions implemented by the state in order to ensure that specific health goals are met. Public health policies range from formal legislation to community outreach efforts [26]. The quality of healthcare provision was viewed under the prism of the well-established Donabedian model, according to which information about optimal care can be drawn from three categories: "structures",

"processes" and "outcomes" [27]. The most important quality indicators in the health sector are divided into three major categories: structural indicators related to facilities, medical equipment and staff, process indicators related to the time it takes to carry out a diagnostic test, waiting times, etc., and outcome indicators linked to disease-specific survival rates and patients' overall satisfaction [28].

### 2.6. Search Strategy

A systematic search was conducted in three electronic databases (PubMed, Scopus and Cumulative Index to Nursing and Allied Health Literature (CINAHL)) for articles published during a 15-year period of time (1 January 2008 to 1 March 2023). Articles were also retrieved from the websites of Orphanet and the National Organization for Rare Disorders (NORD) if they met the PICO and inclusion criteria. The main keywords were: "barriers and facilitators", "parents", "adolescents", "children and young people", "rare disease", "quality of life" and "healthcare provision" combined with the Boolean operators "OR" and "AND". We also used synonyms of the main keywords for the search terms. Keywords and synonyms were applied in the "Title/Abstract" section for the PubMed and Scopus databases, the "MW (Word in subject heading) and all filed" section for the CINAHL database. The search strategy is presented in Supplement Table S1.

This review protocol was prepared using the Preferred Reporting Items for Systematic Reviews and Meta-Analyses Protocol (PRISMA-P) guidelines [29]. The PRISMA 2020 Checklist & PRISMA 2020 for Abstracts Checklist were completed and the protocol was registered on the International Prospective Register of Systematic Reviews (PROSPERO) (ID: CRD42023422686) [30].

### 2.7. Study Screening

The results from the database searches were organized through the Mendeley Reference Manager and duplicates were removed. All titles and abstracts were screened by two reviewers (PT and GK) independently and discrepancies were resolved through discussion with the scientific supervisor (ES). PT subsequently screened all of the full-text copies of the selected articles, as this review is part of a PhD dissertation, and consulted with ESS for final approval. The final number of studies included in the review was 25 [12,13,31–41]. The PRISMA flow diagram describing the search process is presented in Figure 1.

### 2.8. Quality Assessment

The quality assessment of the included studies was performed using the Mixed Methods Appraisal Tool (MMAT; version 2018) [42]. This tool is considered efficient in appraising the methodological quality of different research traditions (qualitative, quantitative and mixed methods studies). The first author (PT) appraised the 25 selected studies [12,13,31–41], deliberating with ESS. The quantitative studies and mixed methods studies had an MMAT rating of 100% each, while two of the qualitative studies' ratings were 80%. The results are presented in Appendix A, Tables A1–A4.

### 2.9. Data Extraction

This review followed a convergent integrated approach for data extraction, which involves data transformation and allows reviewers to combine quantitative and qualitative data [43]. According to JBI typology for MMSR, the process of "qualitizing" was applied, referring to quantitative data being converted into themes or categories [44]. Data extraction was conducted by the first author (PT) after deliberation with ESS.

### 2.10. Data Synthesis

A qualitative reflexive thematic analysis was subsequently conducted whereby the barriers and facilitators within each domain of the ecological model were organized into sub-themes. The thematic analysis followed the six phases of reflexive thematic analysis outlined by Braun and Clarke: (1) familiarization; (2) data coding; (3) generating initial

themes; (4) reviewing and developing themes; (5) refining, defining and naming themes; and (6) writing the report [45]. The main trunk of all steps was conducted by PT and supervised by ESS.

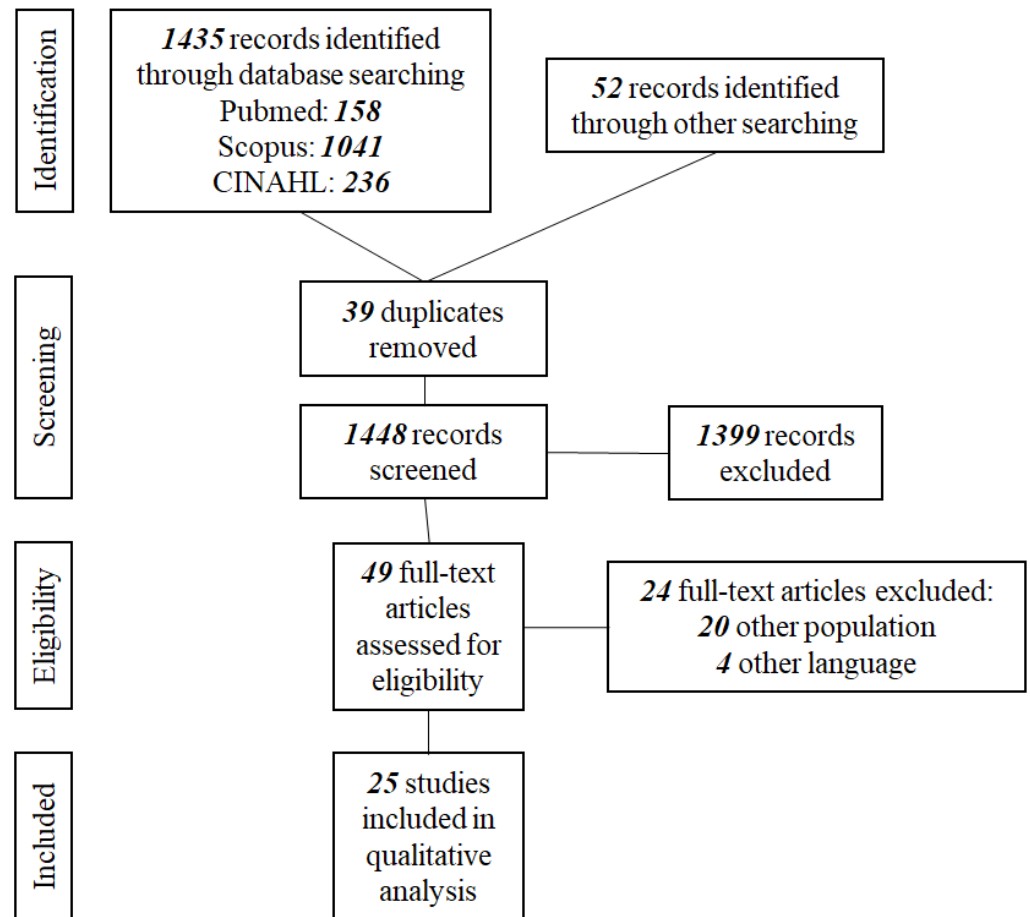

**Figure 1.** PRISMA flow diagram.

## 3. Results

### 3.1. Study Identification Characteristics and Synthesis of Key Findings

Twenty-five articles met the review's inclusion criteria. The identification characteristics of the selected studies [12,13,31–41] (author/year, country, type of study, population of study, tools used and RDs involved) are given in Table 1. Fourteen studies were conducted in Europe [11,31–33,38,40,41,46–52], four studies in Australia [34,35,53,54], three studies in Canada [12,13,39], three studies in the USA [37,55,56] and one study in China [57]. There were seventeen qualitative studies [12,13,31–33,37–41,46,47,49,51,53,55,56], six quantitative studies [11,34,48,50,52,57], and two studies [35,54] were conducted using a mixed method. There was a broad spectrum of RDs involved, which practically represent all human body systems affected (genetic disorders, congenital malformations, metabolic disorders, neurodevelopmental disorders, disorders of the nervous system, blood diseases, gastrointestinal diseases, diseases of skin and subcutaneous tissue, diseases of the eye and its annexes, diseases of the respiratory system, diseases of the genitourinary system and metabolic diseases). The selected articles' key findings are synthesized in Table 2.

**Table 1.** Study characteristics.

| Author (Year) | Country | Type | Participants | Tools | Rare Diseases |
|---|---|---|---|---|---|
| Adama (2021) [34] | Australia | Cross-sectional | Parents (N: 41) | Parent Stigma Scale (PSC)<br>Revised Olweus Bully/Victim Questionnaire (OBVQ)<br>Index of Social Competence (ISC)<br>Strengths and Difficulties Questionnaire (SDQ) | Musculoskeletal diseases, blood/oncology diseases, chromosomes/genetic or congenital diseases, metabolic disorders, nervous system disorders, immune system disorders |
| Anderson (2013) [35] | Australia | Mixed methods | Families (N: 30) | Health Utilities Index Mark II (HUI-II)<br>Impact on Family scale (IOF)<br>Royal Alexandra Hospital for Children Measure of Function (RAHC MOF) | Genetic metabolic disorders |
| Baumbusch (2018) [12] | Canada | Qualitative | Parents (N: 16) | Semi-structured interviews | Not defined |
| Boettcher (2020) [50] | Germany | Cross-sectional | Families (N: 75) | Ulm Quality of Life Inventory for Parents (ULQIE)<br>Brief Symptom Inventory (BSI)<br>Coping Health Inventory for Parents (CHIP)<br>Oslo-Social Support Scale (OSSS-3)<br>Family Assessment Measure (FAM) | Rare Diseases that require mechanical ventilation |
| Bogart (2014) [37] | USA | Qualitative | Adolescents (N: 10) | Eight open-ended questions drawn from previous research | Moebius syndrome |
| Cardinali (2019) [38] | Italy | Qualitative | Parents (N: 15) | Semi-structured interviews | Aicardi syndrome, Angelman syndrome, Arginine succinic aciduria, Chromosome 22 Ring, Fryns syndrome, Goldenhar syndrome, Klinefelter syndrome (49 XXXXY), Lesch–Nyhan syndrome, Mucolipidosis type III, Prader–Willi syndrome, Rett syndrome, Wolf–Hirschhorn syndrome |
| Currie and Szabo (2019) [39] | Canada | Qualitative | Parents (N: 15) | Semi-structured interviews | Genetic diseases, metabolic disorders, nervous system disorders |
| Currie and Szabo (2019) [13] | Canada | Qualitative | Parents (N: 15) | Semi-structured interviews | Neurodevelopmental disorders |
| Damen (2022) [51] | Netherlands | Qualitative | Mother (N: 1) | Personal narrative | Neurofibromatosis type I |
| Gao (2020) [57] | China | Cross-sectional | Parents (N: 651) | Pediatric Quality of Life Inventory™ 4.0 (PedsQL™ 4.0) | Patent ductus arteriosus, infantile agranulocytosis, autoimmune thrombocytopenia, polysyndactyly, Hirschsprung disease, cleft lip and palate, tetralogy of Fallot, myasthenia gravis, Guillain–Barré syndrome, glycogen storage disease, Langerhans cell histiocytosis |
| Geerts (2008) [52] | Netherlands | Cross-sectional | Families (N: 29) | Johns Hopkins Adult Cystic Fibrosis Program Survey (7-item parent version and 6-item patient version)<br>Haemo-QoL-A, versions for adolescents, adults and their parents<br>Parents' illness-related distress (van Dongen-Melman) | Hemophilia |

**Table 1.** *Cont.*

| Author (Year) | Country | Type | Participants | Tools | Rare Diseases |
|---|---|---|---|---|---|
| Gimenez-Lozano (2022) [11] | Spain | Cross-sectional | Families (N: 163) | Semi-structured, self-completed questionnaire with 52 questions | Congenital malformations, genetic disorders, nervous system diseases, metabolic diseases, blood diseases, diseases of the circulatory system, gastrointestinal diseases, disease of the skin and subcutaneous tissue, disease of the eye and its annexes, diseases of the respiratory system, diseases, of the genitourinary system, neurodevelopmental disorders |
| Gómez-Zúñiga (2019) [46] | Spain | Qualitative | Families (N: 10) | Semi-structured interviews | Not defined |
| Hanson (2018) [53] | Australia | Qualitative | Adolescents (N:20) | Semi-structured interviews | Primary lymphoedema |
| Huyard (2009) [47] | France | Qualitative | Parents (N: 15) | Semi-structured interviews | Fragile X syndrome, Cystic fibrosis, Wilson's disease, mastocytosis, locked-in syndrome, very rare syndromes |
| Magliano (2013) [48] | Italy | Cross-sectional | Parents (N: 494) | Barthel index (BI) Family Problems Questionnaire (FPQ) Social Network Questionnaire (SNQ) | Duchenne, Becker, or limb–girdle muscular dystrophies |
| Mazzella (2021) [54] | Australia | Mixed methods | Adolescents (N: 44) | Quality of Life Survey SMA Health Index instrument (SMA-HI) free text response | Spinal muscular atrophy |
| Palacios-Ceña (2018) [49] | Spain | Qualitative | Parents (N: 31) | In-depth interviews, focus-groups, field notes, personal documents | Rett syndrome |
| Pasquini (2021) [55] | USA | Qualitative | Parents (N: 15) | Semi-structured interviews | Metachromatic leukodystrophy and spinal muscular atrophy |
| Sisk (2022) [56] | USA | Qualitative | Parents (N: 24) | Semi-structured interviews | Complex vascular malformation, overgrowth disorders |
| Smits (2022) [40] | Netherlands | Qualitative | Parents (N: 12) | Questionnaires and subsequent interviews | Genetic diseases |
| Somanadhan and Larkin (2016) [41] | Ireland | Qualitative | Parents (N: 8) | In-depth interviews | Mucopolysaccharidosis |
| Verger (2021) [31] | Spain | Qualitative | Parents (N: 8) | In-depth interviews, focus-groups | Not defined |
| Vines (2018) [32] | UK | Qualitative | Adolescents (N: 9) | Semi-structured interviews | Cystic fibrosis |
| Witt (2019) [33] | Germany | Qualitative | Parents (N: 73); Adolescents (N: 47) | Pediatric Quality of Life Inventory™ 4.0 (PedsQL™ 4.0) Short-form 8 Questionnaire (SF-8) | Achondroplasia |

**Table 2.** Synthesis of the key findings of the included studies.

| Study | Synthesis of Key Findings |
| --- | --- |
| Adama (2021) [34] | Parents report high incidence of health-related stigma, bullying and social disorientation for their suffering children. They also refer to limited targeted school-based interventions. **The authors propose strategic development of policies in order to address the specific emotional, behavioral, educational and social needs of adolescents with RDs.** |
| Anderson (2013) [35] | Parents caring for adolescents with RDs are negatively impacted by delays in diagnosis, lack of easy access to peer support groups, low social awareness and excessive financial burdens. **The paper concludes that more analytical studies need to be conducted aiming to optimize healthcare delivery models.** |
| Baumbusch (2018) [12] | Parents refer to limited knowledge from key healthcare providers and impeach the State for major systemic issues regarding access to healthcare. The quality of the services provided is considered "questionable". Parents also suffer from out-of-pocket payments and experience employment difficulties. **The article elaborates on the notion of "expert patient" and "expert caregiver" in the absence of formally designated care coordinators.** |
| Boettcher (2020) [50] | Parents experience feelings of guilt, anger and depression and detach themselves from their parental role. Mothers' mental health is severely compromised since they are traditionally the main caregivers of the sick family member. **The article highlights the need for psychosocial screening and support for parents of children with RDs.** |
| Bogart (2014) [37] | Adolescents with Moebius syndrome describe positive and negative experiences focusing on peer relationships, social engagements and interactions with parents and healthcare personnel. **The article suggests targeted interventions in order to raise social awareness and to empower effective coping mechanisms.** |
| Cardinali (2019) [38] | The protective role of social support for the parents is well established in academic literature. The complexities of caregiving and associated gender differences are further studied and analyzed. **The paper discusses the shortage of structured health policies and the geographical scattering of health institutions for youth with RDs.** |
| Currie and Szabo (2019) [39] | Parents encounter gaps in the accessibility to government support and face the hardships of fragmented and high-cost care. They function as advocates, medical navigators and disease managers. **The article highlights the need for an integrated approach from networks of healthcare and social support providers.** |
| Currie and Szabo (2019) [13] | Parents undergo a sense of silencing and frustration due to their becoming therapists, caregivers and navigators for their medically fragile children. They are also dissatisfied with their interaction with providers. **The authors discuss that a holistic healthcare strategy needs to be developed aiming to address discontinuity of care and bad coordination between health and social services.** |
| Damen (2022) [51] | The paper follows the storytelling of a mother caring for an adolescent with NF1. She illustrates the impact of the management of the disease in between hospitals, organizations and services for every single family member. **The article foregrounds the multilayer burdens in family dynamics when the child suffers from an RD and highlights the deficits of targeted policies.** |
| Gao (2020) [57] | Physical activity, mental health and school performance of patients with RDs should be frequently and carefully monitored. **The authors point out that research is essential on the treatment, production, implementation and availability of orphan drugs.** |
| Geerts (2008) [52] | The transition from pediatric to adult care in the field of RDs is still, in most cases, unorganized and may lead to worsening of the disease and the mental and health deterioration of the parents. **The paper argues in favor of the implementation of policies that facilitate all aspects of transition, taking in account the gender identity of the main caregiver.** |
| Gimenez-Lozano (2022) [11] | Public resources for the multilayer needs of RD patients are limited and frequently do not cover all services required. **The article focuses on deficiencies in the healthcare system and suggests different approaches in regional and national healthcare planning regarding RDs.** |
| Gómez-Zúñiga (2019) [46] | Communication between key health providers and the family of the patient is considered crucial to the effective management of the RD. **The authors propose an adjustment of "mutual trust" between doctors and family members cultivating availability and empathy.** |
| Hanson (2018) [53] | Adolescents with PL struggle with low self-esteem and treatment restrictions. **The paper suggests the development of strategies in order to empower young patients to advocate for themselves in societal context.** |

**Table 2.** *Cont.*

| Study | Synthesis of Key Findings |
| --- | --- |
| Huyard (2009) [47] | Patients' experience with RD would be better if doctors exhibited more respect regarding privacy, diagnosis information and disclosure.<br>**The paper brings forward the need for policies that enable healthcare professionals to meet their patients' moral expectations.** |
| Magliano (2013) [48] | The task of caregiving for a sick child with RD is dependent on family dynamics and influenced by professional and social support. Parents still remain the main caregivers and coordinators of the RD.<br>**The article poses the question of providing models of care that strengthen the parents as advocates and healthcare navigators.** |
| Mazzella (2021) [54] | Young RD patients value the support from healthcare personnel who listen to them and take into account their lived experiences, worries and preferences.<br>**The authors encapsulate adolescents' views on schooling, socialization and disease management and suggest more studies that give prominence to the voice of the patients themselves.** |
| Palacios-Ceña (2018) [49] | Parents of children with Rett syndrome mention obstacles in diagnosis, health processes and financial management of the RD.<br>**The paper indicates more careful planning of health policies, health systems and support policies.** |
| Pasquini (2021) [55] | Health insurance experiences of parents of SMA and MLD patients reveal problematic access to multi-level services required and time consuming conversations with insurance representatives.<br>**The paper underlines the need for formally designated healthcare professionals as official coordinators of the RD.** |
| Sisk (2022) [56] | Parents of children with VM report discrepancies in initial care and maintenance therapeutics for the patients.<br>**The authors provide insights to multiple factors that impede optimal care.** |
| Smits (2022) [40] | RDs are complex chronic conditions that affect young patients and may contribute negatively to family functioning.<br>**Interdisciplinary family-centered models of care are suggested by the authors.** |
| Somanadhan and Larkin (2016) [41] | Parents of young MPS patients comment on the range of uncertainties regarding the RD in everyday life.<br>**The paper highlights the negative impact of frequent relapses and hospitalizations on the quality of life of the whole family.** |
| Verger (2021) [31] | Coordination of healthcare and education is recognized as an area for major improvement.<br>**The article argues that professionals of different fields must have a non-stereotyped approach to young people with RDs.** |
| Vines (2018) [32] | Adolescents suffering from cystic fibrosis experience isolation as part of the disease's everyday care in order to protect themselves from cross-infections. This results in rejection from peer groups, low self-esteem and bio-psychological strain.<br>**The paper suggests an increase in RD knowledge and awareness and relevant targeted policies.** |
| Witt (2019) [33] | Achondroplasia patients and their families suffer from chronic and debilitating consequences.<br>**The paper marks the importance of caring for the psychosocial well-being of the entire family as part of managing the RD.** |

### 3.2. Barriers and Facilitators to Optimal Healthcare for Adolescents with RDs

The socio-ecological model, as first described by Russian Psychologist Bronfenbrenner, is characterized by distinct traceable systems in the life of the studied subjects [58]. The model is widely used in psychology and healthcare studies. Uhm, Choi and Lee utilized the model effectively in two research papers regarding chronic diseases in young people [18,59]. It conceptualizes the idea that systems involving individuals' environments play dimensional roles and frequently overlap. Problems are firstly constructed through their interconnections and then applied strategies are designed.

In Table 3, we code the key findings and identified barriers and facilitators on five levels according to the predefined conceptual framework.

**Table 3.** Barriers and facilitators to optimal care.

| Ecological Model of Health-Levels | Studies | Barriers | Facilitators |
|---|---|---|---|
| **Intrapersonal** | Bogart (2015) [37], Hanson (2018) [53], Magliano (2013) [48], Mazzella (2021) [54], Vines (2018) [32] | Limited knowledge of patients and families<br>Limited self and family education<br>Limited self-esteem and stress-coping mechanisms | Information-sharing between parents and doctors, doctors and patients |
| **Interpersonal** | Anderson (2013) [35], Baumbusch (2018) [12], Boettcher (2020) [50], Bogart (2015) [37], Currie and Szabo (2019) [13], Damen (2022) [51], Gimenez-Lozano (2022) [11], Gómez-Zúñiga (2019) [46], Huyard (2009) [47], Pasquini (2021) [55], Smits (2022) [40], Verger (2021) [31] | Scarce knowledge of healthcare stakeholders (medical and paramedical personnel, community services personnel, insurance representatives and other parties)<br>Lack of awareness of school staff<br>Limited awareness in society groups (schoolmates, peer groups, parents' relatives, employers and colleagues) | Supported self and parental care<br>Patients' dynamic groups and advocacy organizations<br>Rare Disease Awareness and Assistance Programs |
| **Institutional/ Organizational** | Adama (2021) [34], Anderson (2013) [35], Baumbusch (2018) [12], Boettcher (2020) [50], Currie and Szabo (2019) [39], Currie and Szabo (2019) [13], Gao (2020) [57], Geerts (2008) [52], Gimenez-Lozano (2022) [11], Gómez-Zúñiga (2019) [46], Huyard (2009) [47], Mazzella (2021) [54], Palacios-Ceña (2018) [49], Pasquini (2021) [55], Sisk (2022) [56], Smits (2022) [40], Somanadhan and Larkin (2016) [41], Witt (2019) [33] | **Structures**<br>Lack of genetic laboratories, facilities and tests to establish an early diagnosis<br>Geographically distributed health and social services<br>Centers of expertise mainly placed in tertiary levels of care or non-existent<br>National registries complex, incomplete or non-existent<br>Limited industry conducting research on orphan drugs<br>Shortage of expert professionals in all fields (doctors, nurses, speech therapists, ergo therapists, play therapists, psychologists, technicians, etc.)<br>**Processes**<br>Limited RD education in general doctors and primary care personnel<br>Lack of protocols/guidelines for clinical management and follow-up<br>Disconnection between primary, secondary and tertiary level of care<br>Information not effectively shared among all professionals involved<br>**Outcomes**<br>Inequity and inaccessibility in holistic care<br>Dissatisfied parents emotionally and financially burnt out<br>Patients with frequent relapses, hospitalizations, deterioration and unmet needs<br>Mistrust in the healthcare system and the government | **Structures**<br>Rare Disease Centers of Expertise geographically planned<br>Genetic services and counseling accessible<br>**Processes**<br>Well-educated physicians, nurses and other professionals<br>National registries taxonomized for each group of diseases linked with appropriate medications' prescriptions and other treatments<br>Single entry point and official healthcare coordinator<br>**Outcomes**<br>Better services for patients and optimized health outcomes<br>Psychosocial care for patients and families |

**Table 3.** *Cont.*

| Ecological Model of Health-Levels | Studies | Barriers | Facilitators |
|---|---|---|---|
| **Community** | Baumbusch (2018) [12], Gimenez-Lozano (2022) [11], Pasquini (2021) [55], Verger (2021) [31] | Inadequate communication between family members and local authorities<br>Absence of single entry point and official care coordinator<br>School staff uneducated, absence or inadequacy of school nurses, unsafe school environment | Community-based interventions/Rare Disease Assistance Programs<br>Equipped local health services<br>Educated primary healthcare physicians<br>Designation of formal healthcare coordinator<br>Pragmatic healthcare plan<br>Collaboration with school personnel and school nurses |
| **Public and Policy** | All 25 selected studies [12,13,31–41] | Lack of public funding<br>Lack of industry interest due to the scarcity of knowledge and rarity of the disorders<br>Limited university/academic/research funding and programming for RDs<br>Theoretical approaches not linked to actual methods and intervention programs<br>Restricted integration and evaluation policies for healthcare services involved | Public funding<br>Expanding neonatal screening tests<br>Primary healthcare awareness and education<br>Comprehensive healthcare planning starting from first day of diagnosis |

### 3.2.1. Intrapersonal Level

Five studies identified barriers and facilitators at the intrapersonal level as perceived by patients and their parents [32,37,48,53,54].

Barriers

Patients recognize gaps in self-esteem, self-development and self-awareness directly linked to physical disabilities and treatment isolation due to the nature of their RD [37,54]. Experiencing burdens of pain, fatigue and the need for treatment compliance may hamper feelings of self-appreciation and happiness [32,53]. This can result in intrapsychic stress and dysfunctional coping mechanisms both for adolescents and their parents. Adolescents sometimes prioritize personal well-being and socializing and disrupt their medication [53]. Parents admit to high levels of anxiety and frustration due to inadequate knowledge and limited or fragmented psychosocial care [54].

Facilitators

Support from families, friends and schoolmates, and balanced information sharing between the family, school and medical community can empower adolescents and caregivers [54]. Self-care and family support assist adolescents to develop resilience and personal growth and take responsibility for themselves [53].

### 3.2.2. Interpersonal Level

Twelve studies identified barriers and facilitators at the interpersonal level as perceived by patients and their parents [11–13,31,35,37,40,46,47,50,51,55].

Barriers

Parents undergo a sense of silencing and disorientation due to their becoming therapists, caregivers and navigators for their medically fragile children [13]. Many of them feel disconnected from their parental role and demonstrate (a) a high incidence of anxiety, depression, sleep disorders and "chronic sadness" and (b) confined personal time plus the disruption of professional and social relationships [50]. Mothers endure heavier caregiver burdens being traditionally the main caretaker of the afflicted child. Women most often report switching to part-time employment or leaving the workforce in order to attend medical appointments and supervise their child [50,51].

Adolescents and parents are also dissatisfied with their interaction with key healthcare providers, who often exhibit a lack of awareness regarding specific RD implications and stereotyped/prejudiced behaviors [12]. Moreover, children report scarce knowledge from school teachers, classmates and peer groups [31].

Facilitators

Young RD patients value the support from healthcare personnel who listen to them and take into account their worries and preferences [54]. An adjustment of "mutual trust" between doctors and family members cultivating availability and empathy leads to better health outcomes [11]. Targeted school-based interventions can prevent the negative psychosocial outcomes among children with RDs. The "social architecture model" whereby school teachers reconstruct peer groups and rearrange the inclusion of children with RDs is considered positive. Moreover, extra-curricular activities can be tailored upon consultation with the family and the adolescent's doctors in ways that fit the adolescent's participation [34].

### 3.2.3. Institutional/Organizational Level

Seventeen studies identified barriers and facilitators at the institutional/organizational level as perceived by patients and their parents [11–13,33–35,39–41,46,47,49,50,52,54–57]. The barriers and facilitators at this level are examined according to the Donabedian Model of Care.

Structures

Barriers

Parents report a lack of genetic laboratories and tests to establish an early diagnosis [12,35,41,49]. The majority refer to Centers of Expertise or multidisciplinary

clinics, but point out that these are mainly placed in tertiary levels of care, resulting in traveling long distances to obtain standard care [35]. National registries are incomplete or non-existent and electronic records are not systematically reviewed and distributed between all professionals involved [11]. Both Rare Disease-specific coding and mutual use of the relevant information sources are lacking in general practice [49].

Facilitators

Service delivery via appropriately resourced specialized centers with medical staff, pharmacy and access to equipment and information in one location is reported as best practice. Moreover, parents consider that informed patient-held electronic health records improve their experiences when accessing different health professionals [12,38].

Processes

Barriers

Most parents feel that information about support organizations, psychologists and social services should be offered at the time of diagnosis and managed by an officially designated healthcare coordinator [11,12,35,55]. Pharmaceutical industries exhibit little interest in conducting research on orphan drugs and associated delays in health technology assessments result in holding back drug authorization and treatment management [57]. Medication obstacles appear both in diagnosing and in obtaining and continuing the treatment [56]. Associated insurance bureaucracy (obtaining insurance, maintaining insurance, covering all needs and not only hospital healthcare) aggravate the necessary therapeutic approach and burden the parents with out-of-pocket payments [12,55]. Parents report shortages of expert professionals in all fields (doctors, nurses, speech therapists, physical therapists, play therapists, psychologists, technicians, etc.) [11,13,39].

Facilitators

National Rare Disease Registries taxonomized for each group of diseases linked with appropriate medications' prescriptions and other treatments facilitate young patients as to become "visible" and "acknowledged" by the health system [11]. Adolescents and parents value the formation of dynamic Patient Advocacy Groups and the institutionalization of national organizations for RDs as positive. These communities most frequently guide families to psychosocial services that enhance patients' autonomy and improve their social inclusion [33,40,54]. The establishment of an official healthcare coordinator (doctor, nurse or other professional) for all aspects of the disease right from the first day of the diagnosis is considered a milestone towards effective care [12,55].

Outcomes

Barriers

Patients and parents report inequity and inaccessibility in holistic care and families express emotional and financial burnout [12,34,35]. Dissatisfactions are worsened when the sick family member is confronted with frequent relapses, hospitalizations, deterioration and lack of systematic psychosocial support [50]. The transition from pediatric to adult healthcare is often unrecognized and does not follow an individualized pragmatic care plan. This causes uncertainty and poses the danger of neglecting therapy and aggravating the disease [52].

Facilitators

Self-support groups, formal healthcare coordinators, a well-informed team of providers and orphan drug availability are related to better health outcomes [11,35,47,55]. Surveys that use validated tools to assess the impact of policies on the health functioning of the afflicted child are encouraged by families, as they make them feel "heard" and not "silent and neglected" [11,57].

3.2.4. Community Level

Four studies identified the barriers and facilitators at the community level as perceived by patients and their parents [11,12,31,55].

Barriers

Parents report insufficient training of primary care professionals, who are often the first contact point of the family and who usually undertake the chronic follow up [11,55]. Parents also complain of disconnections between services, even among professionals of the same service, including bad information transfer in public administrations [31].

Facilitators

Primary healthcare centers incorporating family-centered care and community-based aspects can play a crucial role in coordinating health and social care and utilizing appropriate computer technology [46]. Peer-support resources, starting in the community and emerging to social media platforms, enable adolescents to accept themselves and cope with the disease. Parent-to-parent interaction provides the families with a sense of unity and mutual support [12].

### 3.2.5. Public and Policy Level

Barriers

Parents complain of a lack of public funding, limited pharmaceutical interest due to the scarcity of knowledge and rarity of the disorders, and limited academic/research funding and programming for RDs [11,13,33,35,38,39,41,47,49,54,57]. Most of the countries in the included studies have elaborated on national policies for RDs and prioritize them within their health ministries. Nevertheless, most of the selected studies eventually underline the need to transform national and regional health public plans into effective practice with a measurable impact on patients and families. The redistribution of the public budget according to their standardized needs should also be organized [11,37,50–53].

Facilitators

The factors that enhance family satisfaction are policies to allocate more public resources, to expand neonatal screening and to incorporate genetic counselors and psychologists into primary healthcare. The evaluation of the integrative services with follow-up periods and outcome measures should include (1) adolescent health outcomes, (2) health and social service use, (3) healthcare quality indicators, (4) school absenteeism and education issues and (5) cost-effectiveness [11,12,41,54,55].

## 4. Discussion

Rare Diseases (RDs) in young people are characterized by low frequency and great heterogeneity, are usually chronic and degenerative, and cause physical disabilities and psychological burden. Their treatment requires services at multiple levels (from home and community care to tertiary hospitals) and should be multi-disciplinary (pediatricians, geneticists, psychologists, occupational therapists, speech therapists, physiotherapists, etc.), so the coordination of the different services involved is also required [60]. However, few studies emphasize the inadequacy of integrated care practices as a key element in improving the family's quality of life when the adolescent suffers from an RD [14,61]. This mixed systematic review used the ecological model of health in order to identify the barriers and facilitators to optimal healthcare provision as perceived by adolescents with RDs and their parents. Patients and their parents, regardless of age and individual diagnosis, share common experiences and collectively recognize barriers resulting in unmet health needs.

In general, unmet needs included delayed, incorrect or missing diagnoses. Delays and denials were detected both in diagnostic processes as well as in insurance-covered treatments, approved medications, and investigational and off-label therapeutic agents [62,63]. Genetic diagnosis and genetic counseling services are almost prohibited for families of low socioeconomic status or ethnic and racial minorities [57]. Furthermore, we detected a lack of specialized information and a shortage of qualified health professionals. A number of studies point to the need for RD education, initiating in medical schools and expanding to primary care physicians and general practitioners [64–66]. According to the World Health Organization, primary healthcare (PHC) functions as a network of cooperation between medical and local communities [67]. PHC staff should become the intersectional regulator

of triage and correct guidance for adolescents with RDs as patient-users in the health system, depending on their needs and preferences. Furthermore, both RD-specific coding and the use of the relevant RD information sources are lacking in general practice. This results in miscalculations in the epidemiology of RDs and the underestimation of necessary financial resources. In addition, the missing information perplexes and may even prohibit scientific knowledge [68].

Our review led to the identification of additional challenges including inequalities and difficulties in accessing medical, social, and psychological care [11,31]. Previous studies have shown that these are directly linked to both issues with diagnosis and quality healthcare services as patients and families most frequently circulate between health facilities and different professionals until the RD is identified [69]. Even when the diagnosis is established, continuous and effective monitoring of the RD is negatively dependent on the geographical scattering of the required services and the disconnection between them [70,71].

Catastrophic health costs also constitute a significant problem. Parents continue to feel "stressed, overwhelmed and overextended" with caregiving at home, while similar care in medical facilities requires expertise and expensive providers [54]. Health insurances do not fully cover the adolescent's social and medical needs and discussions with insurance representatives are often frustrating and misleading [55]. All of the above may result in the loss of confidence in the health system and associated policies. In parallel to other studies, our review elaborates on the notion of "expert patients" and "expert caregivers" becoming navigators, advocators and managers of the RD [5,11,35,72–74]. On the other hand, parents feel that larger influences and industry circles constrain their voices, and one of the key messages in our review is that further research into personal narratives from adolescents and parents is needed [13,75].

All of the 25 selected studies [12,13,31–41] recognize more barriers than facilitators in optimal care and pinpoint the most important inadequacies at the organizational level. The institutionalization of integrated care towards RDs seems to be the answer to the abovementioned unfulfilled needs, but significant obstacles are described by patients and their families. This review highlighted six major optimal care barriers: (a) the absence of a single entry point, (b) the lack of a formal care coordinator, (c) geographically dispersed health and social services, (d) the lack of an individualized pragmatic care plan incorporating transition to adult services, (e) psychosocial care not being routinely provided and (f) theoretical policies not being linked to active intervention programs. The single entry point and official case manager has been previously proposed to specific care programs both in the European Union (EU) and the United States in small-scale populations with positive results [76]. The establishment of formal care coordinators can extenuate potential healthcare access inequities, as the ability to navigate healthcare systems varies according to the socio-structural conditions experienced by families [12]. The care coordinator joins different services and relates health appointments into a wider plan and should preferably work in the community/primary healthcare facilities, close to the family [39]. Our conclusions line up with other studies that highlight the inconsistency between strategic planning at three systemic levels (national, state and local) and corresponding practices [77]. First, the centralized model of integrating social services in specialized reference centers is expensive for the state. Second, the linkage model is considered difficult to operate due to the rarity and heterogeneity of diseases. The third model, that of care coordination, although considered by stakeholders as the best option for RDs, has rarely been implemented and has been pilot tested mainly in adult and mental health patients [78]. Few studies have developed taxonomy of care coordination for RDs [61,77] and a small number of actual family-based intervention programs are still not evaluated for their impact on patients' welfare [79].

Adolescence is a special period of life during which bargaining self-identity and building independence are vital issues. Therefore, the handling of a debilitating disease, which may implicate physical deformities, mental strains and psychological distresses, is

challenging and requires structured supportive mechanisms [37]. Our review elucidated that health-related stigma, bullying and social disorientation are perceived as great barriers in the optimal care of the affected adolescents. This is in accordance with previous studies showing that despite treatment improvements, everyday normalcy and school acceptance still tend to be stalemated [80]. Most of the included studies attribute the current societal impasse to the scarcity of targeted protocols in schools and other fields of extracurricular activities [31]. Parents acknowledge and teachers agree that the lack of clearly identified procedures for information sharing between school staff and healthcare professionals constitute obstacles in the adolescent's everyday functioning [81]. Adolescents seek more community-based campaigns to enhance awareness in peer groups and targeted protocols for their participation in athletic and leisure activities [34].

Finally, our review pointed out the following factors as major facilitators to effective care: first, National and Local Rare Disorders Organizations and Patient Advocacy Groups. These communities manage to raise awareness, educate medical and societal circles, organize congresses and present a dynamic "wave" to provide high-quality information for patients and support for their families [82]. In many cases, Patient Advocacy Groups recommend specific treatments to regulatory bodies, contact genetic counselors, assist with clinical trials, and make crucial suggestions for laws, legislation and action plans in disease management [12,83,84]; second, empathetic and amiable scientists ready to ensure that the whole well-being of the family constitutes an important goal in the management of the disease. Fluent communication with a team of providers (physicians, nurses, social workers and other therapists) helps parents maintain family integration, understand the medical situation and develop coping mechanisms [36,39]. Optimizing electronic health record functionality and the distribution of data between key care providers is perceived as a boost for caregivers and patients. Telehealth can also be used for urgent medical consultations and appointments [72]; and third, politicians and governors who actually "care" and are willing to commit to solutions with and for the patients [85].

## 5. Limitations

Our review followed the constructed steps of a mixed methods systematic review and has some limitations. The included studies come from European countries, Australia, USA, Canada and China, so the results should be viewed in this context and consider cultural bias. Moreover, the included studies were conducted in three research traditions, involved different types of RDs and assessed factors from a mixed sample of patients and organizations. In order to address these limitations, the thematic analysis focused on searching and generating main common themes concerning RDs in adolescents. The barriers and facilitators were methodically categorized in order to ultimately represent universal healthcare concerns transferable to all RDs. Nevertheless, we would like to point out that patients may benefit from more detailed studies targeted to specific RD diagnoses.

## 6. Conclusions

According to Donabedian, raising awareness among society and healthcare professionals about system design and planning is important, but not enough. In the case of Rare Diseases, healthcare systems are challenged to provide multi-targeted medical and psychosocial care with no ethnic or individual prejudices. The moral and cultural sensitization of the involved professionals can be an important mechanism of mobilization, but implementing specific interventions from a central authority is the cornerstone for system improvement.

To our knowledge this is the first review attempting to provide a systematic snapshot of the barriers and facilitators in optimizing care provided to adolescents with Rare Diseases, as perceived by the patients themselves and their parents. The independent barriers and facilitators were organized into five levels of healthcare, though our review shows that specific dimensions intercross and a shortage of policies at one level is proportionally related to all levels. More detailed adolescent and parental studies are required in order to enhance their voices and communicate their concerns to health policymakers.

**Supplementary Materials:** The following supporting information can be downloaded at: https://www.mdpi.com/article/10.3390/pediatric15030043/s1, Table S1: PRISMA 2020 Checklist. Ref. [86] in Supplementary Materials part.

**Author Contributions:** Conceptualization, P.T.; methodology, P.T.; validation, P.T. and G.K.; investigation, P.T.; data curation, P.T. and G.K.; writing—original draft preparation, P.T.; writing—review and editing, P.T., G.K. and E.S.S.; visualization, P.T.; supervision, E.S.S.; project administration, P.T. All authors have read and agreed to the published version of the manuscript.

**Funding:** This research received no external funding.

**Institutional Review Board Statement:** Not applicable.

**Informed Consent Statement:** Not applicable.

**Data Availability Statement:** All data are presented in the article. The corresponding author may be contacted for further queries.

**Conflicts of Interest:** The authors declare no conflict of interest.

**Appendix A**

**Table A1.** Search strategy.

| Database | Key Words |
| --- | --- |
| PubMed | ((((((rare disease*[Title/Abstract]) OR (orphan disease*[Title/Abstract])) OR (rare disorder*[Title/Abstract])) OR (orphan disorder*[Title/Abstract])) AND ((((((barriers[Title/Abstract]) OR (obstacles[Title/Abstract])) OR (difficulties[Title/Abstract])) OR (facilitators[Title/Abstract])) OR (enablers[Title/Abstract]))) AND ((((((healthcare provision[Title/Abstract]) OR (structures[Title/Abstract])) OR (processes[Title/Abstract])) OR (outcomes[Title/Abstract])) OR (healthcare settings[Title/Abstract])) |
| Scopus | (TITLE-ABS-KEY ("rare diseases" OR "orphan diseases" OR "rare disorders" OR "orphan disorders") AND TITLE-ABS-KEY (barriers OR obstacles OR difficulties OR facilitators OR enablers) AND TITLE-ABS-KEY ("healthcare provision" OR structures OR processes OR outcomes OR "healthcare settings")) |
| CINAHL | (("rare diseases" or "orphan diseases" or "rare disorders" or "orphan disorders") and (barriers or obstacles or difficulties or facilitators or enablers) and ("healthcare provision" or structures or processes or outcomes or "healthcare settings")) |

**Table A2.** Quality appraisal of qualitative studies using MMAT 18.

| Study | Is the Qualitative Approach Appropriate to Answer the Research Question? | Are the Qualitative Data Collection Methods Adequate to Address the Research Question? | Are the Findings Adequately Derived from the Data? | Is the Interpretation of Results Sufficiently Substantiated by Data? | Is There Coherence between Qualitative Data Sources, Collection, Analysis and Interpretation? |
| --- | --- | --- | --- | --- | --- |
| Baumbusch (2018) [12] | Y | Y | Y | Y | Y |
| Bogart (2014) [37] | Y | Y | Y | Y | Y |
| Cardinali (2019) [38] | Y | Y | Y | Y | Y |
| Currie and Szabo (2019) [39] | Y | Y | Y | Y | Y |
| Currie and Szabo (2019) [13] | Y | Y | Y | Y | Y |
| Damen (2022) [51] | Y | Y | Y | Y | Y |
| Gómez-Zúñiga (2019) [46] | Y | Y | Y | Y | Y |
| Hanson (2018) [53] | Y | Y | Y | Y | Y |
| Huyard (2009) [47] | Y | Y | Y | Y | Y |
| Palacios-Ceña (2018) [49] | Y | Y | Y | Y | Y |
| Pasquini (2021) [55] | Y | Y | Y | Y | Y |
| Sisk (2022) [56] | Y | Y | Y | Y | Y |
| Smits (2022) [40] | Y | Y | Y | Y | Y |
| Somanadhan and Larkin (2016) [41] | Y | Y | Y | Y | Y |
| Verger (2021) [31] | Y | Y | Y | CT | Y |
| Vines (2018) [32] | Y | Y | Y | CT | Y |
| Witt (2019) [33] | Y | Y | Y | Y | Y |

Abbreviations: CT, cannot tell; Y, yes.

**Table A3.** Quality appraisal of quantitative studies using MMAT 18.

| Study | Are the Participants Representative of the Target Population? | Are Measurements Appropriate Regarding Both the Outcome and Intervention (or Exposure)? | Are There Complete Outcome Data? | Are the Confounders Accounted for in the Design and Analysis? | During the Study Period, Is the Intervention Administered (or Exposure Occurred) as Intended? |
|---|---|---|---|---|---|
| Adama (2021) [34] | Y | Y | Y | Y | Y |
| Boettcher (2020) [50] | Y | Y | Y | Y | Y |
| Gao (2020) [57] | Y | Y | Y | Y | Y |
| Geerts (2008) [52] | Y | Y | Y | Y | Y |
| Gimenez-Lozano (2022) [11] | Y | Y | Y | Y | Y |
| Magliano (2013) [48] | Y | Y | Y | Y | Y |

Abbreviation: Y, yes.

**Table A4.** Quality appraisal of mixed method studies using MMAT 18.

| Study | Is There an Adequate Rationale for Using a Mixed Methods Design to Address the Research Question? | Are the Different Components of the Study Effectively Integrated to Answer the Research Question? | Are the Outputs of the Integration of Qualitative and Quantitative Components Adequately Interpreted? | Are Divergences and Inconsistencies between Quantitative and Qualitative Results Adequately Addressed? | Do the Different Components of the Study Adhere to the Quality Criteria of Each Tradition of the Methods Involved? |
|---|---|---|---|---|---|
| Anderson (2013) [35] | Y | Y | Y | Y | Y |
| Mazzella (2021) [54] | Y | Y | Y | Y | Y |

Abbreviation: Y, yes.

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
