# Peer review of "Barriers to and Facilitators of Providing Care for Adolescents Suffering from Rare Diseases: A Mixed Systematic Review"

_pediatrrep, doi:10.3390/pediatric15030043_

Round 1

Reviewer 1 Report

Dear Authors,

congratulations for the great work you did to review this very intriguing is topic.. The multidisciplinary approach is essential as you wrote in the introduction. Could you define better which range of age do you mean for adolescent? and about age of tansition. Could you also explain better the methods you used as Bach-Motensen, ecc.

The management of adolescent patients is peculiar and heterogeneous for any different disorder (genetic, metabolic, endocrinological, haematological, ecc) in terms of unmet needs and also for many countries with different Health systems. We suggest to try to compact the table 1 and 2 that result too dispersive and also table A2 .We appreciate a lot the table 3 that is very usefull to understand the literature. The interpersonal level 3.22 touches a crucial point. The importance to have a psychologist during follow-up visit is a great option of tailor-made unmet need. And the transition is another great point to face at international level and in terms of patients association and advocacy. There is a global poor acknowledgment for rare diseases with low investment. The discussion is well definte and conducted.

If possible, it could be usefull to summarize the 2 levels: barriers and facilitators. Telemedicine could be a great solution and also individualization of needs of patients.

only minor revision

Author Response

Reviewer 1:

Point 1: Could you define better which range of age do you mean for adolescent? And about age of transition. Could you also explain better the methods you used as Bach-Motensen, ect.

Response 1: We thank the reviewer for taking the time to make this useful comment. The review employed the World Health Organization’s (WHO’s) definition for adolescence: “the phase of life between childhood and adulthood, from ages 10 to 19” as stated in the text in 2.4. Nevertheless, following your suggestion, we added an extra definition of adolescence and specifically referred to the age of transition in 2.4. Identity issues of adolescence and lack of structured transition protocols are further analysed in Discussion. We followed the PRISMA protocol for systematic reviews and the JBI typology for Mixed Methods Systematic Reviews. We believe that clear explanation of the procedure is given in 2.5, 2.6, 2.7, 2.8, 2.9 and 2.10. As you recommended, we have better described the Bach-Mortensen approach to Mixed Methods Systematic Reviews as follows:

2.4. Definitions

According to Bach-Mortensen and Verboom [19] most mixed methods studies do not explicitly define the terms “barriers and facilitators” and this may reflect negatively on mapping the importance of these factors. Therefore, in this review we identified “barriers and facilitators” as used by the Integrated Checklist of Determinants of Practice (ICDP): “being factors that might prevent (barrier) or enable (facilitator) improvements in healthcare provision for the targeted population” [20]. According to Bach-Mortensen and Verboom “barriers and facilitators” studies should also clearly state the way specific factors are extricated and subsequently amalgamated as well as describe all the steps towards this synthesis [19].

The review employed the World Health Organization’s (WHO’s) definitions for rare diseases (RDs): “diseases with a frequency of less than 6.5 to 10 per 10.000 people” [21] and adolescence: “the phase of life between childhood and adulthood, from ages 10 to 19” [22]. Adolescence is the period between 10-19 years old. It’s a unique era of human development where multifaceted physical, mental and cognitive evolution occurs and the time during which healthy hygiene and self-care patterns are predicated and evolved in adult life. Transition from childhood to adult services for individuals with chronic diseases usually takes place between 16 to 19 years old, that is during middle and late adolescence and is highly dependent on the nature of the healthcare system and the family’s preferences [22,23].

Point 2: We suggest to try to compact the table 1 and 2 that result too dispersive and also table A2.

Response 2: We thank the reviewer for this comment. We have considered the above suggestion, however we find it very difficult to consolidate these tables since the number of our studies is 25 and the study characteristics and basic key findings cannot be omitted or suppressed without compromising the quality of this review. Table A2 refers to the quality appraisal of the studies and is also lengthy due to the number of the papers included in our review.

Point 3: If possible, it could be useful to summarize the 2 levels: barriers and facilitators.

Response 3: We acknowledge the above comment. We respectfully clarify that both barriers and facilitators are summarized separately for each one of the five levels of the socio-ecological model of health in paragraph 3.2 in a way we think is adequately comprehensive for the readers. Moreover, in Table 3 we also provide a snapshot of barriers and facilitators for each level of care.

Reviewer 2 Report

Thanks the authors for completing the search on this topic, and writing up this systematic review.

With reference to similar papers previously published, I would suggest to include more databases for a more comprehensive systematic review.

e.g.

Uhm JY, Choi MY. Barriers to and Facilitators of School Health Care for Students with Chronic Disease as Perceived by Their Parents: A Mixed Systematic Review. Healthcare (Basel). 2020 Nov 21;8(4):506. doi: 10.3390/healthcare8040506. PMID: 33233468; PMCID: PMC7712821.

https://pubmed.ncbi.nlm.nih.gov/33233468/

They used PubMed, Cumulative Index to Nursing and Allied Health Literature (CINAHL), Embase, and Web of Science (WOS) as databases for the search. I believe they are more comprehensive than just es PubMed, Scopus
and CINAHL.

Figure 1 has unmatched figures. Pubmed 158 + Scopus 1,041 + CINAHL 236 = 1,435, which is different from what the authors have shown in Figure 1 down the flow chart. Please explain the reason behind and the details of the 100 studies omitted in the analysis.

Author Response

Point 1: With reference to similar papers previously published, I would suggest to include more databases for a more comprehensive systematic review.

e.g.Uhm JY, Choi MY. Barriers to and Facilitators of School Health Care for Students with Chronic Disease as Perceived by Their Parents: A Mixed Systematic Review. Healthcare (Basel). 2020 Nov 21;8(4):506. doi: 10.3390/healthcare8040506. PMID: 33233468; PMCID: PMC7712821. They used PubMed, Cumulative Index to Nursing and Allied Health Literature (CINAHL), Embase, and Web of Science (WOS) as databases for the search. I believe they are more comprehensive than just PubMed, Scopus and CINAHL.

Response 1: We thank the reviewer for taking the time to make this useful comment. Apart from Pub Med, Scopus and CINAHL we have also searched the websites of Orphanet and the National Organization for Rare Disorders (NORD) in order to present a full review targetted to Rare Diseases as stated in 2.6. Following the reviewer’s comment we conducted a manual search in Web of Science, which did not elicitate any additional relevant results. Therefore, we are confident that our systematic review covers a comprehensive overview of the current literature on adolescent rare diseases.

Point 2. Figure 1 has unmatched figures. Pubmed 158 + Scopus 1,041 + CINAHL 236 = 1,435, which is different from what the authors have shown in Figure 1 down the flow chart. Please explain the reason behind and the details of the 100 studies omitted in the analysis.

Response 2: We thank the reviewer for this useful comment. This was a typographical error on our part. We have now corrected the numbers in Figure 1 to represent the accurate subsets of the flow chart. Thank you.